# *NpPP2-B10*, an F-Box-Nictaba Gene, Promotes Plant Growth and Resistance to Black Shank Disease Incited by *Phytophthora nicotianae* in *Nicotiana tabacum*

**DOI:** 10.3390/ijms24087353

**Published:** 2023-04-16

**Authors:** Guo Wen, Zhongyi Xie, Yao Yang, Yuxue Yang, Qigao Guo, Guolu Liang, Jiangbo Dang

**Affiliations:** 1Key Laboratory of Horticulture Science for Southern Mountains Regions of Ministry of Education, College of Horticulture and Landscape Architecture, Southwest University, Chongqing 400715, China; guowen2020@126.com (G.W.); xzy960526@163.com (Z.X.); yangyao0814@126.com (Y.Y.); yyx111321@163.com (Y.Y.); qgguo@126.com (Q.G.); 2Academy of Agricultural Sciences of Southwest University, State Cultivation Base of Crop Stress Biology for Southern Mountainous Land of Southwest University, Chongqing 400715, China; 3College of Horticulture, China Agricultural University, Beijing 100193, China; 4College of Biological Sciences, China Agricultural University, Beijing 100193, China

**Keywords:** *Nicotiana tabacum*, F-box-Nictaba, lectin, *Nicotiana plumbaginifolia*, black shank

## Abstract

Black shank, a devastating disease affecting tobacco production worldwide, is caused by *Phytophthora nicotianae*. However, few genes related to *Phytophthora* resistance have been reported in tobacco. Here, we identified *NpPP2-B10*, a gene strongly induced by *P. nicotianae* race 0, with a conserved F-box motif and Nictaba (tobacco lectin) domain, in the highly resistant tobacco species *Nicotiana plumbaginifolia*. *NpPP2-B10* is a typical F-box-Nictaba gene. When it was transferred into the black shank-susceptible tobacco cultivar ‘Honghua Dajinyuan’, it was found to promote resistance to black shank disease. *NpPP2-B10* was induced by salicylic acid, and some resistance-related genes (*NtPR1*, *NtPR2*, *NtCHN50*, and *NtPAL*) and resistance-related enzymes (catalase and peroxidase) were significantly upregulated in the overexpression lines after infection with *P. nicotianae*. Furthermore, we showed that *NpPP2-B10* actively regulated the tobacco seed germination rate, growth rate, and plant height. The erythrocyte coagulation test of purified NpPP2-B10 protein showed that NpPP2-B10 had plant lectin activity, and the lectin content in the overexpression lines was significantly higher than that in the WT, which could lead to accelerated growth and improved resistance of tobacco. SKP1 is an adaptor protein of the E3 ubiquitin ligase SKP1, Cullin, F-box (SCF) complex. We demonstrated that NpPP2-B10 could interact with the NpSKP1-1A gene in vivo and in vitro through yeast two-hybrid (Y2H) and bimolecular fluorescence complementation (BiFC), indicating that NpPP2-B10 likely participates in the plant immune response by mediating the ubiquitin protease pathway. In conclusion, our study provides some important insights concerning *NpPP2-B10-*mediated regulation of tobacco growth and resistance.

## 1. Introduction

Tobacco black shank is a devastating disease caused by *Phytophthora nicotianae,* which harms the roots, stems, and leaves of tobacco at various growth stages, causing water-like disease lesions, yellowing, and even death of the plants [1]. Black shank-resistant cultivated tobacco germplasm sources are scarce, and only the resistant cultivars ‘Florida 301’ [2] and ‘Beinhart 1000’ [3] are commonly mentioned. Nevertheless, some wild species in the genus *Nicotiana,* such as *N. rustica*, *N. longiflora*, and *N. plumbaginifolia,* are highly resistant to black shank [4,5]. Their resistance has been successfully transferred to cultivated tobacco by distant hybridization [6]. A study by Goins and Apple [7] showed that the resistance of *N. longiflora* and *N. plumbaginifolia* is regulated by the major genes *Phl* and *Php*, respectively, and the black shank resistance was introgressed into cultivated tobacco as early as 1960 [8,9]. In recent years, researchers have identified the glutathione S-transferase gene in *N. tabacum* as a negative regulator, and silencing of this gene has been found to promote resistance to black shank in tobacco [10]. NtPIP, a pathogen-inducible protein, is a positive regulator of tobacco susceptibility to *P. nicotianae* [11]. Apart from the genes mentioned above, few genes related to tobacco black shank resistance have been reported. Therefore, there is a need to mine more genes that can positively regulate tobacco resistance to black shank disease and growth, and such efforts are of great significance for tobacco disease resistance breeding.

Previous studies have revealed that *P. nicotianae* secretes toxins in the host that cause the host cells to breakdown and produce colloids, which block the movement of water in the roots and cause tobacco wilting [12]. Some researchers have proposed that the secretions of this pathogen may contain polygalacturonase, which can decompose the middle lamella and vessel wall of host cells, producing fragments of pectin products that then form gum-like substances [13]. Wang believed that the toxin secreted was a glycoprotein [14]. Therefore, research on tobacco resistance to *P. nicotianae* should focus on some of the subtle molecular mechanisms that link tobacco to the glycoproteins secreted by *P. nicotianae*.

Recent studies have shown that lectins, as immune receptors and/or defense proteins, play an important role in the natural immune system of plants [15,16]. Lectins are proteins with at least one carbohydrate-binding site that can selectively recognize specific polysaccharides. This allows lectins to recognize pathogen invasion by binding to exogenous sugars, thus triggering downstream immune defense responses. Cell wall glycoproteins, named cellulose-binding elicitor lectin (CBEL), have been found localized in the cell wall of *P. nicotianae* mycelium and play a key role as effectors during the plant-pathogen recognition process [17]. In addition, lectins are abundant proteins in seeds and vegetative tissues and were originally thought to be storage proteins that provide nitrogen sources for plants and promote plant growth and development [18]. Studies have shown that lectins can store, package, and transport nutrients to protect seeds during seed maturation and germination [19]. Lectins can also be involved in the development of plant embryo cells as mitogenic factors to regulate cell division, differentiation, growth, and development. Thus, lectins can be used to develop cultivars with better growth and development under adverse conditions. Lectins are very diverse, each binding to a specific sugar [16]. Nictaba (*N. tabacum* agglutinin) was first discovered in *N. tabacum* and is induced by MeJA treatment, and it specifically binds to N-acetylglucosamine [20]. The phloem protein (PP2) family is considered a subgroup of the Nictaba family [21]. In Arabidopsis, two Nictaba domain proteins, AN4 and AN5, significantly increased tolerance to *Pseudomonas syringae* [22]. The phloem-secreted protein, PP2, of *Cucurbita maxima* binds bacteria and fungi to the cross-linked filaments, sealing off the injured phloem sieve tubes from further infection by pathogens [23]. In addition, Lee also identified the protein AtPP2-1A in Arabidopsis, which has inhibitory effects on a variety of fungi [24]. Some researchers have proposed that PP2 proteins play a role in plant defense against pathogens by directly binding to chitin cell walls and promoting wound healing through the formation of self-assembling filaments [25].

Some Nictaba N-termini are characterized by an F-box motif called F-box-Nictaba-type proteins. The F-box motif usually consists of 40–50 amino acid residues. F-box proteins bind to SKP1, Cullin, and RBX proteins to form an E3 ubiquitin ligase, called the SKP1, Cullin, F-box (SCF) complex, which enables specific recognition of degraded substrate proteins [26]. F-box-Nictaba protein has been reported to interact with the core structures of high mannose and complex N-glycosylase in Arabidopsis and rice [27]. Since F-box-Nictaba has both the ability of F-box proteins and the ability to bind to plant carbohydrates, it is speculated that F-box-Nictaba has the function of glycoprotein degradation, similar to the mammalian FBS protein [27]. An F-box-Nictaba gene in Arabidopsis was reported to enhance resistance to *P. syringae* [28]. According to previous studies, F-box-Nictaba genes are most likely to play a role in the innate immune response of plants. However, the F-box-Nictaba proteins in tobacco have not been implicated in the molecular function of plant disease resistance. Tobacco has always been severely affected by black shank disease, and there is no evidence related to Phytophthora resistance in the study of F-box-Nictaba proteins in other species. Therefore, it is particularly important to further elucidate the role of F-box-Nictaba genes in tobacco disease resistance.

In this study, we identified the F-box-Nictaba gene *NpPP2-B10* in *N. plumbaginifolia* and overexpressed it in the black shank-susceptible tobacco cultivar ‘Honghua Dajinyuan’. We found that *NpPP2-B10* can improve the resistance of tobacco to *P. nicotianae* race 0 and could positively regulate the growth and development of tobacco. This is of great significance to the agricultural production of tobacco. Furthermore, our study revealed that NpPP2-B10 possesses both F-box protein properties and plant lectin activity, suggesting that NpPP2-B10 has the ability to bind polysaccharides and participate in protein degradation. In conclusion, our results provide an important reference for further understanding the nature of black shank resistance in *N. plumbaginifolia*.

## 2. Results

### 2.1. Characterization of NpPP2-B10

In previous studies, we analyzed the differentially expressed genes of the highly resistant tobacco species *N. plumbaginifolia* and of the moderately resistant cultivar ‘Yunyan 87’ identified by transcriptome sequencing [9]. A Unigene, c62451.graph_c0, was specifically expressed in *N. plumbaginifolia.* In further studies, it was found that the c62451.graph_c0 was rapidly induced 6–72 h after infection with *P. nicotianae* race 0, which was verified by using real-time quantitative polymerase chain reaction (RT-qPCR). The expression level of the c62451.graph_c0 was the highest at 72 h after infection, at which point it was 15 times higher than that in the control (Figure 1a), suggesting that the gene responds to the infection of *P. nicotianae* race 0.

Plant hormones play an important role in the expression and regulation of plant defense genes and responses to biotic and abiotic stresses. Salicylic acid (SA), jasmonic acid (MeJA), and ethylene (ETH) are important signaling molecules in plants that can activate downstream defense-related genes [29]. The expression of c62451.graph_c0 in *N. plumbaginifolia* was significantly higher than that in the control group at 6–72 h after SA treatment, and the expression level was 8.5 times higher than that in the control group after 48 h (Figure 1b). In addition, c62451.graph_c0 was also induced by MeJA and inhibited by ETH (Appendix A), but c624511.graph_c0 was more responsive to SA than these two hormones. These results suggest that c62451.graph_c0 may be regulated by the SA signaling pathway.

To further investigate the function of this gene, we conducted a BLAST search in NCBI using the partial sequence. Based on the BLAST results, we cloned the gene and named it *NpPP2-B10* (GenBank: OM264753). The total length of the coding sequence (CDS) of the gene was 786 bp (Appendix A), the open reading frame (ORF) encoded 261 amino acids, and the calculated molecular weight (MW) was 29,643.71. Conserved domain analysis revealed that the N-terminus of the gene contained an F-box protein motif, and that the C-terminus contained a PP2 structure (Nictaba-related lectin domain), which had an additional F-box motif that common Nictaba lacks (Figure 1c). Therefore, NpPP2-B10 can be classified as an F-box-Nictaba class protein. In previous studies, two tryptophan residues in the Nictaba domain (Trp 15 and Trp 22) were considered key to the binding of tobacco lectin to carbohydrates [30]. Similarly, these two Trp residues are also highly conserved in NpPP2-B10. This suggests that NpPP2-B10 has potential lectin activity.

To examine the subcellular location of *NpPP2-B10*, we constructed a 35S::*NpPP2-B10*-eGFP fusion expression vector controlled by the CaMV35S promoter and used *Agrobacterium tumefaciens* for transient expression in the leaves of *N. benthamiana*. After 48–72 h of dark culture, the transgenic leaves were stained with 4,6-diamidino-2-phenylindole (DAPI) and observed under a fluorescence microscope. As shown in Figure 1d, green fluorescence was observed in the nucleus and cytoplasm of tobacco transferred into the 35S::*NpPP2-B10*-GFP recombinant plasmid. These results indicate that *NpPP2-B10* is a nuclear and cytoplasmic localized gene, which is consistent with the subcellular localization results of other Nictaba-related proteins [26].

### 2.2. Overexpression of NpPP2-B10 Promoted Seed Germination and Plant Growth in Tobacco

To investigate the function of *NpPP2-B10*, we constructed a 35S::*NpPP2-B10* expression vector and then transformed it into the black shank-susceptible variety ‘Honghuadajinyuan’. Nine 35S::*NpPP2-B10* transgenic lines were obtained by kanamycin screening and PCR identification (Appendix A). Three self-crossing T1 seedlings with the highest expression levels were used to study the phenotypes of the transgenic plants. We made the self-crossing progenies of the three transgenic tobacco lines and wild-type (WT) seeds grow on Murashige and Skoog (MS) medium and observed that the germination of seeds from the transgenic lines was better than that of the WT seeds one week after seeding (Figure 2a). We seeded the *NpPP2-B10*-OE lines and the WT on wet filter paper and recorded their germination rates over the course of one week. Statistical results showed that the seed germination rate of the *NpPP2-B10*-overexpressing lines was significantly higher than that of WT in 4–7 days after seeding (Figure 2b). Then, we transferred the germinated seedlings to sterilized soil for growth, and it was observed that the seedlings of the overexpression lines were larger than the WT seedlings (Appendix A). At the bolting stage, we observed that the height of plants in the *NpPP2-B10*-OE lines was higher than that of WT plants (Figure 2c). We calculated the height of T1 generation plants from the *NpPP2-B10*-OE lines and the height of the WT offspring, and the results showed that the height of the plants in the *NpPP2-B10*-OE lines was significantly higher than that of the WT plants (Figure 2d). These results suggest that the expression of *NpPP2-B10* in tobacco promotes plant growth and development. Lectins are abundant proteins in seeds and vegetative tissues and are considered storage proteins that provide nitrogen sources for plants [18]. We hypothesize that the increased lectin content in *NpPP2-B10*-OE lines caused this phenotype.

### 2.3. Overexpression of NpPP2-B10 Enhanced Tobacco Resistance to Black Shank Disease

Based on previous studies on plant F-box-Nictaba proteins, we speculated that *NpPP2-B10* might play a role in plant disease resistance. We selected five-week-old seedlings of the *NpPP2-B10*-OE lines and WT at the same growth stage and treated them with the *P. nicotianae* race 0 spore suspension by means of root irrigation. Three days after root irrigation, tissue necrosis and decay appeared in the roots and stems of the WT seedlings, while no corresponding symptoms appeared in the stems and roots of the *NpPP2-B10*-OE lines (Figure 3a). At four to nine days after root irrigation, we observed that both the WT and *NpPP2-B10*-OE lines showed symptoms of black shank disease. As shown in Figure 3b, the incidence of infection in the *NpPP2-B10*-OE lines was lower than that in the WT. The statistical results showed that the OE-3 line seedlings and OE-12 line seedlings had a significantly lower incidence of infection than the WT seedlings during the period from four to nine days after root irrigation, and that the OE-44 line seedlings (the line of seedlings with the lowest *NpPP2-B10*-OE expression) had a significantly lower incidence of infection than the WT seedlings only at four dpi (Figure 3c). These results indicated that *NpPP2-B10* positively regulates tobacco resistance at the seedling stage.

In the field, tobacco black shank mainly harms the stems and roots of adult plants. Therefore, we further observed the root phenotype of tobacco infected with *P. nicotianae* race 0 in mature plants. Due to the higher lignin content in adult tobacco, both the *NpPP2-B10*-OE lines and the WT had enhanced resistance to black shank, so the phenotype of *NpPP2-B10*-OE lines at this time was not as obvious as it was at the seedling stage. Two days after pathogen infection, we observed the development of lesions on the stems of the tobacco plants, and the area of the lesions on the *NpPP2-B10*-OE plants was significantly lower than that on the WT plants (Figure 3d). The resistance of the OE-44 plants was still lower than that of OE-3 and OE-12 plants, which was consistent with the results from the root irrigation experiment. These results indicate that the *NpPP2-B10* gene positively regulates tobacco resistance to black shank disease at the mature stage.

### 2.4. Validation of Lectin Activity of the NpPP2-B10 Protein

Based on the existence of conserved carbohydrate-binding sites (Trp 15 and Trp 22) in NpPP2-B10, we speculated that NpPP2-B10 would have potential lectin activity. To verify this conjecture, we constructed the prokaryotic expression vector pET-SUMO-*NpPP2-B10*. The purified NpPP2-B10 protein was obtained by prokaryotic expression (Figure 4b). One percent mouse red blood cells and two percent rabbit red blood cells were used to validate the lectin activity of the protein, and the red blood cells, the target protein, and physiological saline were proportionally mixed and allowed to sit for half an hour. If the target protein had lectin activity, it would combine with polysaccharides on the surface of red blood cells, thereby causing agglutination between cells, and erythrocyte sedimentation would not occur. Instead, red blood cells from the lectin-free wells would settle to the bottom of the V-shaped well [31]. The results showed that the red blood cells agglutinated and did not precipitate in the wells added with NpPP2-B10 protein and in the positive control (ConA) (Figure 4a). The results of gradient dilution showed that at least 31 µg/mL of NpPP2-B10 could agglutinate 1% of mouse erythrocytes, and at least 15.5 µg/mL of NpPP2-B10 could agglutinate 2% of rabbit erythrocytes. This indicates that the specificity of NpPP2-B10 for rabbit erythrocytes was higher than that for mouse erythrocytes. Taken together, these results suggest that the NpPP2-B10 protein possesses lectin activity.

During the tobacco seedling stage, we selected leaves from the *NpPP2-B10*-OE plants and the WT plants to measure the lectin content. The results showed that the lectin content of the *NpPP2-B10*-OE lines was significantly higher than that of WT (Figure 4c). The OE-3 line had the highest lectin content, 1.6 times that in the WT. These results indicate that the NpPP2-B10 protein has lectin activity and was successfully overexpressed in the *NpPP2-B10*-OE lines.

### 2.5. Interaction between NpPP2-B10 and NpSKP1-1A

In Arabidopsis, wheat, rice, and other crops, the SKP1 protein has been shown to be the joint protein of the SCF complex in E3 ubiquitin ligase. The interaction with SKP1 in yeast has been used to determine that F-box proteins are a component of the SCF complex [32]. To determine whether *NpPP2-B10* plays a role as part of the SCF complex, we screened two genes annotated as SKP1 family genes, C74462.graph_c0 (*NpSKP1-1A*) and C85975.graph_c0 (*NpSKP1-21*), from *N. plumbaginifolia* transcriptomes by using qPCR. These two genes responded to black shank disease infection. The expression level of C74462.graph_c0 in the infected plants was 73 times higher than that in the uninfected controls at 6 h after infection, and the expression level of C85975.graph_c0 was 9 times higher in the infected plants than in the uninfected controls at 6 h after infection (Figure 5a,b). Then, we cloned these two genes (Appendix A) and named them *NpSKP1-1A* (OM264752) and *NpSKP1-21* (OM264751), respectively. Analysis of the conserved domains of both showed that NpSKP1-1A had conserved Cullin-binding sites and F-box protein-binding sites (Figure 5c), while NpSKP1-21 did not (Appendix A).

We fused *NpPP2-B10* into the GAL 4-DNA-binding domain and *NpSKP1-1A* and *NpSKP1-21* into the GAL 4-active domain for the yeast two-hybrid experiment, and the results are shown in Figure 5d. NpPP2-B10 interacts with NpSKP1-1A but does not interact with NpSKP1-21 (Figure 5d). We linked *NpPP2-B10* and *NpSKP1-1A* to the vectors SPYNE173 and pSPYCE, respectively, and successfully constructed the expression vectors SPYNE173-*NpPP2-B10* and pSPYCE-*NpSKP1-1A*. Then, *A. tumefaciens* carrying two recombinant vectors was mixed and injected into tobacco plants and cultured for 48–72 h for fluorescence observation. The results showed that there was no fluorescence in the negative control, but fluorescence could be seen in the nucleus and cytoplasm in the experimental group (Figure 5e). These results indicate that NpSKP1-1A interacts with the NpPP2-B10 protein in vitro and in vivo, suggesting that NpPP2-B10 may mediate the ubiquitin protease pathway to regulate tobacco resistance to black shank disease.

### 2.6. NpPP2-B10 Enhanced the Expression of Disease Resistance-Related Genes and the Activity of Disease Resistance-Related Enzymes

*NtPR1*, *NtPR2,* and *NtCHN50* are genes induced by SA and are related to disease resistance [33]. The enzymes catalase (CAT), peroxidase (POD), and phenylalanine ammonium lyase (PAL) have been reported to be related to plant disease resistance [34,35]. To further study the downstream disease resistance-related pathways triggered by *NpPP2-B10*, we detected the expression levels of disease resistance-related genes and the activities of disease resistance-related enzymes in tobacco after pathogen irrigation. The RT-qPCR results showed that the SA induction-related genes *NtPR1*, *NtPR2*, *NtCHN50,* and *NtPAL* were upregulated in the tobacco plants after infection and that the expression levels of the above four genes in the *NpPP2-B10*-OE lines were significantly higher than those in the WT at 3 dpi and 7 dpi (Figure 6a, Appendix A). Of the four genes, the *NtPR1* gene had the strongest response to pathogen infection, with the response in the OE-12 line being 43 times higher than the response in the WT at 7 dpi (Figure 6a). The results of the enzyme activity detection showed that CAT enzyme activity in tobacco first increased and then decreased from 0 to 96 hpi, that CAT enzyme activity in the *NpPP2-B10-*OE line was significantly higher than that in WT at 12, 24, and 48 hpi, and that CAT enzyme activity in the OE-3 line was 6 times higher than that in the WT at 12 hpi (Figure 6b). However, the POD and PAL enzymes showed no significant change in patterns compared to the CAT enzyme. The POD activity of the *NpPP2-B10*-OE line was significantly higher than that in the WT at 6 hpi, 12 hpi, 24 hpi, and 48 hpi, while the PAL activity in the *NpPP2-B10*-OE line was significantly higher than that in the WT only at 6 hpi and 72 hpi (Appendix A).

## 3. Discussion

Unlike most Nictaba-related lectin genes, *NpPP2-B10* carries an F-box motif at its N-terminus, which may result in functional differences [27]. Nictaba lectin was first discovered in *N. tabacum* and is normally unexpressed in leaves, but it is induced by MeJA treatment [20]. However, when *N. plumbaginifolia* was treated with MeJA, the expression of *NpPP2-B10* first decreased and then increased, which was different from the expression pattern of Nictaba lectin of *N. tabacum*. Furthermore, *NpPP2-B10* was strongly induced from 0 h to 72 h after SA treatment in *N. plumbaginifolia*, and the expression level was the highest at 72 h. This is consistent with the expression pattern of other F-box families [36]. For example, the *TaPP2-A13* gene of wheat was induced by SA treatment, and the expression level was the highest at 24 h after SA treatment [37]. These results reflect that the expression of F-box family genes is closely related to salicylic acid.

The subcellular localization of *NpPP2-B10* was in the nucleus and cytoplasm, which may be related to the specific binding between the lectin translated by the *NpPP2-B10* gene and the glycoproteins in the cytoplasm and nucleus. For example, Nictaba binds to the O-GlcNAc unit of histone in *N. tabacum*, thus indirectly changing the transcription of some genes [38]. This may also be related to the F-box characteristics of the *NpPP2-B10* gene. The F-box protein of maize mainly exists in the cytoplasm (37%), chloroplast (28%), and nucleus (25%) [39]. In wheat, 803 (79.3%) F-box proteins were only located in the nucleus, while 132 (13.0%) F-box proteins seemed to have dual or multiple loci [40]. This may be related to the fact that the F-box protein-mediated ubiquitin protease pathway is involved in the degradation of various proteins in plant cells.

We induced the overexpression of the *NpPP2-B10* gene in the susceptible variety ‘Honghua Dajinyuan’ and found that it could significantly improve the resistance of tobacco to black shank disease. In studies on the pathogenesis of citrus Huanglong disease, phloem protein (PP2) has been shown to bind with filamentous protein to exacerbate duct blockage, resulting in blockage of the vessel [41], which leads to the onset of citrus Huanglong disease. However, in this study, the overexpression of the *NpPP2-B10* gene promoted tobacco resistance to black shank. This is likely related to the F-box protein properties (mediated ubiquitin protease pathway) of NpPP2-B10. To investigate this further, we assessed the lectin activity of NpPP2-B10 by using an erythrocyte coagulation assay, and the interaction between NpPP2-B10 and the hypothesized SKP1 family protein NpSKP1-1A in vitro and in vivo was assessed by using a yeast two-hybrid assay and a BiFC assay. These results indicate that NpPP2-B10 possesses both F-box protein and lectin properties. It is speculated that this plant carbohydrate-binding F-box protein may have a similar glycoprotein degradation function as the mammalian FBS protein [27]. Tobacco black shank is a disease caused by the oomycete *Phytophthora* that mainly harms the roots and stems of tobacco in the field. In previous studies, the pathogenic factor has been speculated to be the toxin produced by *P. parasitica*, and researchers currently believe that the toxin is a glycoprotein [14]. We hypothesized that glycoproteins secreted by pathogens were degraded by the glycoprotein degradation function of the NpPP2-B10 protein, which led to the enhancement of tobacco resistance, and we will carry out further studies to test this hypothesis in the future.

Due to the anti-insect/bacterial/fungal/virus functions of plant lectins in nature, it has been speculated that plant lectins may be part of plant immunity through their action of binding to glycoproteins [42], but the specific regulatory mechanism has not been explained. In our study, *NpPP2-B10* was strongly induced by SA, and the expression levels of the SA-dependent defense pathway-related genes *NtPR1* and *NtCHN50* in the *NpPP2-B10-*OE lines were significantly higher than those in the WT after infection with *P. nicotianae* race 0. Similar results were found in a study of the F-box Nictaba protein in *A. thaliana*, wherein overexpression of this gene led to increased resistance and the expression of the *PR1* gene significantly increased after pathogen infection [43]. Therefore, we hypothesized that the *NpPP2-B10* gene may affect plant immune function by participating in the regulation of the SA pathway.

Interestingly, in contrast to most plant defense-related genes, the *NpPP2-B10* gene also promoted the growth and development of tobacco, mainly showing that *NpPP2-B10*-OE lines were superior to the WT in terms of the seed germination rate, growth rate, and plant height. Previously, researchers overexpressed the fungal lectin CCL2 in *A. thaliana* and found that it positively affected plant growth, mainly showing that the fresh and dry weights of CCL2-overexpressing plants were higher than that of WT plants [44]. However, the researchers did not explain the reasons for the positive effects of CCL2 on plant growth and development in greater detail. Plant lectins, as storage proteins, could provide nutrients for seeds. We speculated that the lectin activity of NpPP2-B10 promotes plant growth, because we detected a higher lectin content in the *NpPP2-B10-*OE lines than in the WT.

In our study, we demonstrated that *NpPP2-B10* promoted tobacco resistance to black shank disease and enhanced tobacco growth and development. Meanwhile, we also demonstrated that the NpPP2-B10 protein has F-box protein characteristics and lectin activity. We speculated that this is related to tobacco resistance, but the specific regulatory mechanism remains to be studied. Future studies can be carried out in two areas. One is to verify whether NpPP2-B10 protein can degrade exogenous glycoproteins of *P. nicotianae* race 0 to explain the direct cause of the reduction in the tobacco black shank disease spot area. The second area of focus is to investigate the substrate targeted by NpPP2-B10 to further promote the specific mechanism of the *NpPP2-B10*-mediated ubiquitin protease pathway in regulating resistance.

Based on our findings, we hypothesized a putative model for *NpPP2-B10* regulation of tobacco resistance (Figure 7). In the normal environment, NpPP2-B10 interacts with SKP1 proteins to form the SCF complex, which degrades an unidentified negative regulator through the ubiquitin—proteasome system (UPS) pathway and then maintains the normal expression of some plant disease resistance-related proteins and enzymes (such as PR1, PR2, PAL, and CAT), which maintains tobacco resistance. At the same time, due to the tobacco lectin characteristics of NpPP2-B10, it may regulate plant resistance by recognizing glycoproteins of exogenous fungi.

## 4. Materials and Methods

### 4.1. Plant Materials and Growth Conditions

The following materials were selected for this study: the resistant tobacco species *N. plumbaginifolia*, the tobacco variety ‘Honghua Dajinyuan’, *N. benthamiana*, and three independent T1 lines of transgenic tobacco.

The seeds of *N. plumbaginifolia* were placed on wet filter paper and treated with 100 µM of GA_3_ to break seed dormancy [45]. After germination, the seeds were transferred to sterilized soil. The seeds of the other tobacco types were directly sown in sterilized soil. All the above tobacco types were cultured in a humidity-controlled environment (16 h light/8 h dark cycles, 24 °C). *N. plumbaginifolia* was used for gene cloning and expression analysis under various treatments, ‘Honghua Dajingyuan’ was used for the transgenic test, *N. benthamiana* was used for transient expression, and the transgenic tobacco lines were used for phenotypic analysis.

### 4.2. Pathogen Infection and Hormone Treatment

*P. nicotianae* race 0 spore suspension with a concentration of 1.4 × 10^3^ /mL was prepared according to the previously described methods [46]: The spores, SA (2 mM), MeJA (1 mM), and ETH (50 mM) were applied to 2-month-old tobacco seedlings, and a mixture of 0.1% KNO_3_, 0.1% ethanol, and H_2_O was applied to controls. Samples were taken at 0, 6, 12, 18, 24, 36, 48, and 72 h after treatment, and the relative expression level of the *NpPP2-B10* gene was measured by qRT-PCR.

### 4.3. RNA Extraction, Reverse Transcription, and qPCR

Total RNA was extracted from the tobacco tissues using an RNA extraction kit (DP441, TIANGEN, Beijing, China). First-strand cDNA was synthesized by using a PrimeScriptTM RT reagent Kit with gDNA Eraser (RR047A, TAKARA, Shiga, Japan).

qRT-PCR was conducted to detect changes in gene expression by using qTOWER3 G (Analytik Jena, Jena, Germany). The PCR amplification procedure was as follows: 1 cycle of 95 °C for 30 s, followed by 40 cycles of 95 °C for 20 s, and 60 °C for 1 min. The system was a Novo Startfi SYBR qPCR SuperMix Plus 20 µL system (E096-01A, Novoprotein, Shanghai, China). The 2^−ΔΔCt^ method was used to calculate the relative expression values [47], and *NpEF-1a* and *NtEF-1a* were used as controls for *N. plumbaginifolia* and cultivated tobacco, respectively. All the primers for qRT-PCR are shown in Appendix A.

### 4.4. Overexpression of NpPP2-B10 in Cultivated Tobacco

Using a seamless cloning technique, the *NpPP2-B10* gene was linked to the vector pCAMBIA2300 with the CaMV 35S promoter, and through this, the recombinant plasmid pCAMBIA2300 *NpPP2-B10*-OCS was obtained. Referring to a previous method [46], the constructed recombinant plasmid was transferred into the *A. tumefaciens* LBA4404 strain and used to infect the ‘Honghua Dajinyuan’ tobacco plants by means of the leaf-disc method. After callus differentiation and tissue culture, regenerated plants with kanamycin resistance were ultimately obtained. The tobacco total DNA was extracted by using the improved CTAB method [48], and positive plants were verified by means of specific PCR amplification of the *NpPP2-B10* gene. The relative expression level of the *NpPP2-B10* gene was measured by qRT-PCR. The same method was used to detect positive seedlings from the self-crossing 1 generation of the *NpPP2-B10* transgenic line. The primers are shown in Appendix A.

### 4.5. E. coli Expression and Purification of the Recombinant Protein

The *NpPP2-B10* gene was linked to the bacterial expression vector PET-SUMO by using seamless cloning technology, and the recombinant plasmid was transferred into *Escherichia coli* (DE3). After colony PCR and sequence verification, the purified plasmid was transformed into *E. coli* BL 21(DE3) for protein production. The primers are shown in Appendix A.

Protein extraction and purification were conducted as previously described [49]. Isopropyl β-D-thiogalactoside (IPTG) inducer was added to induce protein expression to a final concentration of 0.1 mM, and the bacteria were cultured for 8 h at 30 °C with shaking. Lysozyme was added to a final concentration of 0.1 mg/mL after the bacteria were collected, and the bacteria were lysed by ultrasonication. Then, 10 µL of the supernatant and precipitate were analyzed by SDS-PAGE. The protein solution was placed in a dialysis bag and dialyzed with NTA buffer for 48 h (4 °C) at a flow rate of 1 mL/min. The column was washed with NTA-0 buffer solution (pH 8.0) until no protein was found in the effluent. Imidazole elution was performed, and the eluent was collected in stages. The column was washed with three times the volume of deionized water and sealed with 20% ethanol. The collected eluent was concentrated by dialysis, and 12 µL was taken for SDS-PAGE analysis.

### 4.6. Detection of Lectin Activity and Content

Then, 50 µL of 0.9% NaCl normal saline was successively added to the V-shaped 96-well plate from left to right, and 50 µL of lectin protein solution (500 µg/mL) was added to the first well and evenly mixed. The protein concentration in the remaining wells was successively diluted by a gradient of 2 times with each successive well. The last well was treated with 0.9% NaCl normal saline as a negative control. Then, 50 µL of the corresponding red blood cells was added to each well, thoroughly mixed, and allowed to stand for half an hour to enable the agglutination of red blood cells to be observed. The rationale for this technique was that when the protein has lectin activity, it causes red blood cells to agglutinate into a network that evenly spreads throughout the well, such that the red blood cells do not precipitate. However, the red blood cells in the negative control wells will all precipitate to the bottom of wells, enabling observation of the red spots at the bottom of the wells.

Fresh tobacco leaf tissue was thoroughly ground in liquid nitrogen, and then extracts of 9 times the sample volume (pH 7.4, PBS) were prepared and centrifuged for 30 min (8000 rpm, 4 °C), and the supernatant was collected and temporarily stored at 4 °C for future use. A plant lectin quantitative detection kit (RX1400205PL, Ruixin, Quanzhou, China) was used to detect the tobacco lectin content. The absorbance (OD value) was measured at a wavelength of 450 nm with a microplate reader. The standard concentration was taken as the ordinate (6 standard wells, plus 1 zero well, and a total of 7 concentration points), and the corresponding absorbance (OD value) was taken as the abscissa. Computer software was used to fit the absorbance with a four-parameter logistic curve (4-PL). The standard curve equation was established, and the OD value was used to calculate the concentration of the sample.

### 4.7. Yeast Two-Hybrid Assay and BiFC

The *NpPP2-B10* gene was linked to the vector pGBKT7, and *NpSKP1-1A* and *NpSKP1-21* were linked to the plasmid pGADT7 by using seamless cloning technology. Then, 50 µL of Y2HGold cells was placed in an ice-water mixture, and 1–3 µg of precooled recombinant plasmid, 5 µL of carrier DNA (prepared at 95–100 °C for 5 min, followed by a rapid ice bath, and then this process was repeated), and 250 µL of PEG/LiAc were successively added, and the sample was pipetted and pumped several times to mix. The mixture was then placed in a water bath at 30 °C for 30 min, and then placed in a 42 °C water bath for 20 min. After centrifugation for 40 s (5000 rpm), the supernatant was removed. The bacteria were resuspended in 400 µL of ddH_2_O, and the supernatant was removed after centrifugation for 1 min. The bacteria were suspended in 50 µL of ddH_2_O, coated onto the dual-deficient screening medium (SD/-Trp/-Leu), and cultured upside down in a 28 °C incubator for 48–72 h after the bacterial solution had completely dried. Several single colonies were randomly selected on multi-deficient screening medium (SD/-Trp/-Leu/-His/-Ade), and the nutrient-deficient plates were divided into two conditions (with or without X-α-Gal) and cultured inverted in the dark at 28 °C for 3–5 days. Color was observed, recorded, and photographed. The primers are shown in Appendix A.

*NpPP2-B10* and *NpSKP1-1A* were cloned into the pSPYCE and pSPYNE vectors by a seamless cloning technique, and then the two recombinant plasmids were transformed into the *A. tumefaciens* (GV3101) strain. The bacterial solution (OD = 0.1) containing two recombinant plasmids was mixed in equal volumes and injected into the lower epidermis of the flat leaves of *N. Benthamiana*. After 48–72 h of dark culture, the epidermis was torn off, and the fluorescence signal was captured with an Observer DP80 fluorescence microscope (Olympus, Tokyo, Japan). The primers are shown in Appendix A.

### 4.8. Detection of Disease Resistance-Related Enzyme Activities

The tobacco POD content was determined by using the guaiacol method [50]. CAT content was determined according to another previously described method [51], and PAL activity was determined using a PAL activity detection kit (BC0210, Solarbia, Beijing, China).

## Figures and Tables

**Figure 1 ijms-24-07353-f001:**
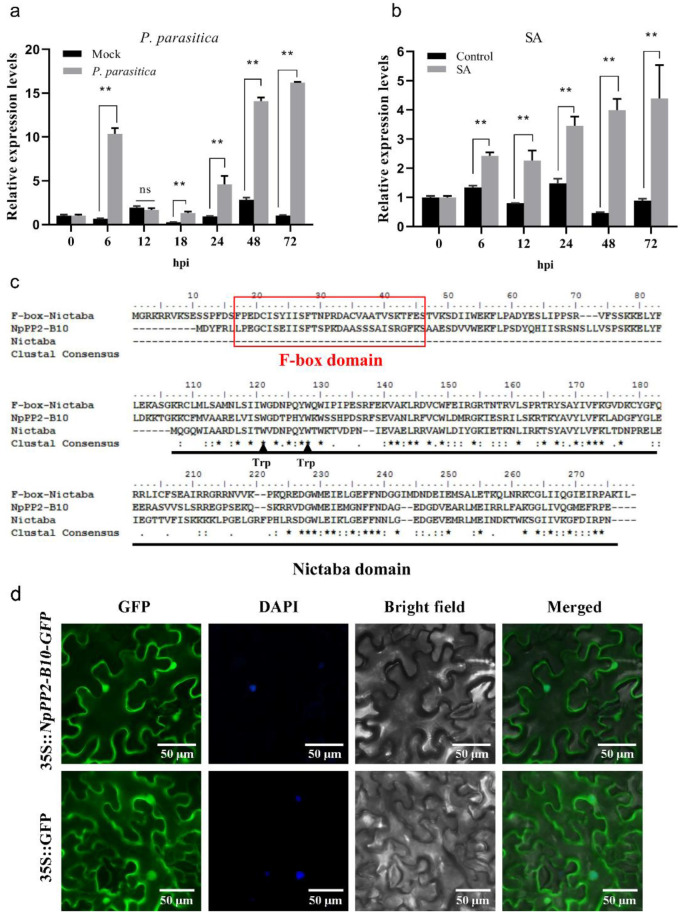
Characterization of *NpPP2-B10*. (**a**,**b**) Expression pattern of the c62451.graph_c0 gene in tobacco treated with *P. nicotianae* race 0 and SA. Asterisks denote significant differences (compared with the control group): ** *p* < 0.01, by Student’s *t*-test; ns, no significant difference. (**c**) Sequence alignment of *NpPP2-B10* with the F-box-Nictaba protein of *Arabidopsis thaliana* and the Nictaba protein of *N. tabacum.* ★ The exact same residue; : residues that are particularly similar in nature; **.** residues that are slightly similar in nature. (**d**) The subcellular localization of *NpPP2-B10* in *N. benthamiana* epidermal cells. GFP: Green fluorescent protein. DAPI: Nucleic marker. Merged: Merged image of the bright field, GFP, and DAPI results. The above image is a random field of view taken from a fluorescent region of the tobacco epidermis.

**Figure 2 ijms-24-07353-f002:**
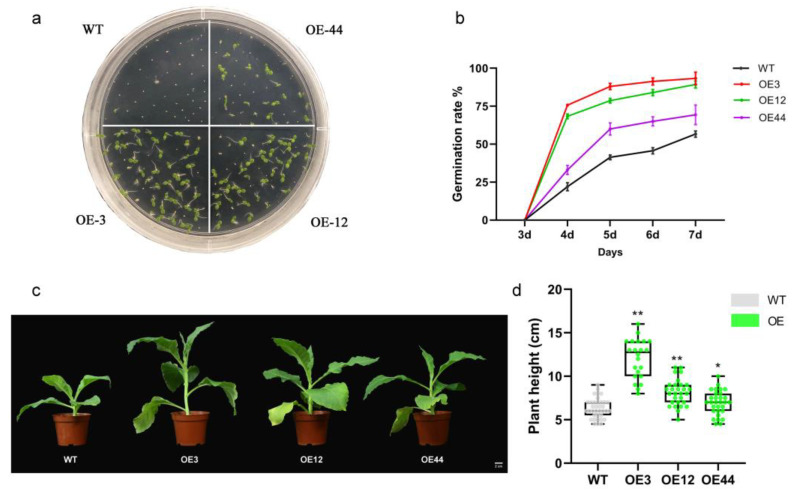
Growth phenotypes of the *NpPP2-B10*-OE lines and WT plants. (**a**) Germination of seeds from each line after one week of incubation on MS medium. (**b**) Germination rate of each line on wet filter paper. (**c**) Growth phenotype of each line at two months of age. (**d**) Plant height of each line at two months of age. Asterisks denote significant differences (compared with the control group): * *p* < 0.05, ** *p* < 0.01, by Student’s *t*-test.

**Figure 3 ijms-24-07353-f003:**
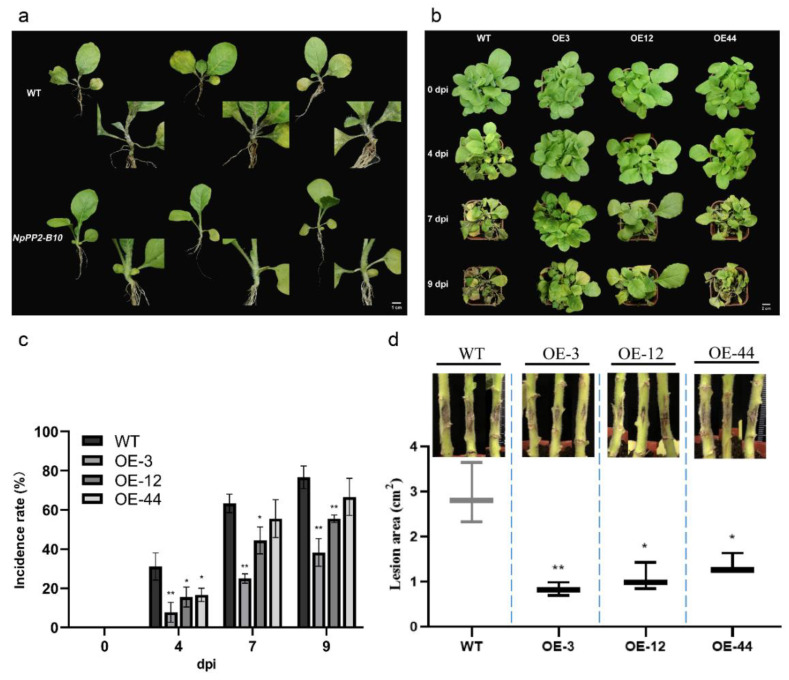
Phenotypes of *NpPP2-B10*-OE lines and the WT after infection with *P. nicotianae* race 0. (**a**) Phenotype of roots of each line 3 days after infection. (**b**) Overall phenotype of each line from 0 to 9 days after infection. (**c**) Incidence rate of each line from 0 to 9 days after infection. (**d**) Phenotype and spot area on the stem of each line after infection. Asterisks denote significant differences (compared with the control group): * *p* < 0.05, ** *p* < 0.01, by Student’s *t*-test.

**Figure 4 ijms-24-07353-f004:**
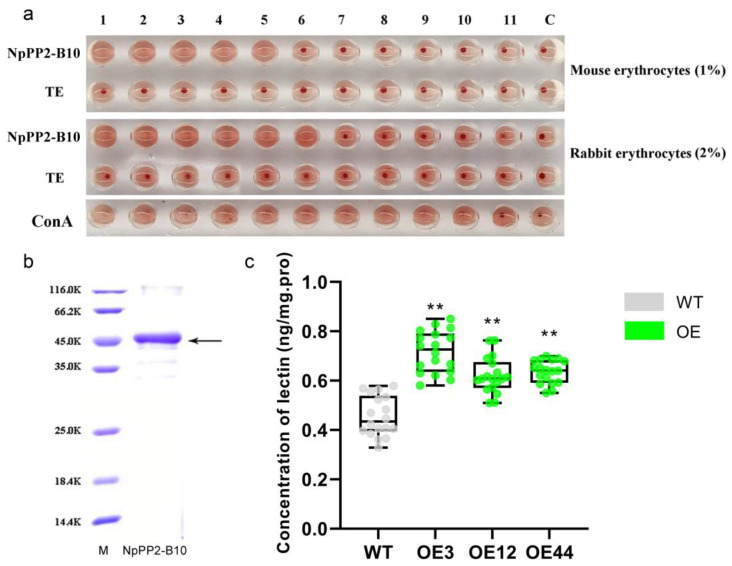
Functional validation of the NpPP2-B10 protein. (**a**) Validation of NpPP2-B10 protein lectin activity. TE: solvent for the NpPP2-B10 protein, as a negative control; ConA: Concanavalin protein with lectin activity, as a positive control. (**b**) Purified NpPP2-B10 protein was detected by means of SDS-PAGE. (**c**) Lectin content in the *NpPP2-B10*-OE and the WT lines. Asterisks denote significant differences (compared with the control group): ** *p* < 0.01, by Student’s *t*-test.

**Figure 5 ijms-24-07353-f005:**
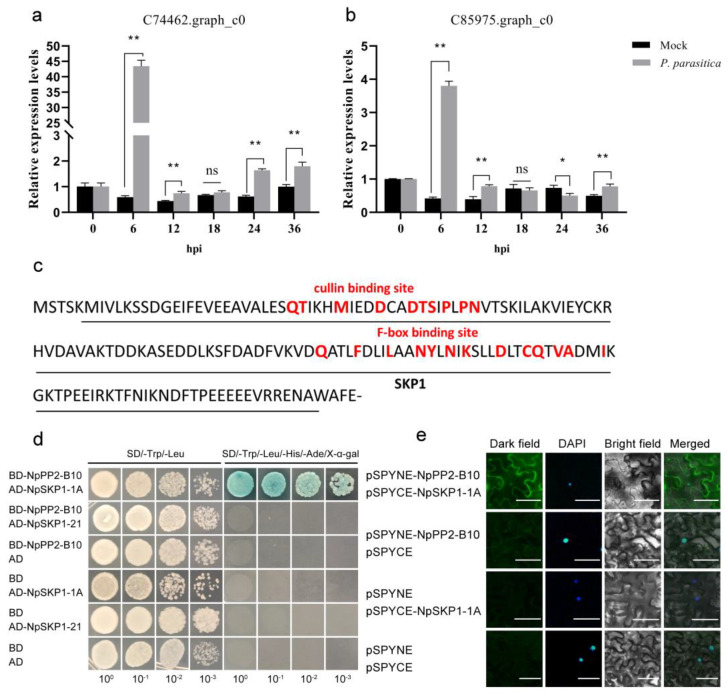
(**a**,**b**) Relative expression levels of c74462.graph_c0 and 85975.graph_c0 after infection with *P. nicotianae* race 0. Asterisks denote significant differences (compared with the control group): * *p* < 0.05, ** *p* < 0.01, by Student’s *t*-test; ns, no significant difference. (**c**) Amino acid sequence of the *NpSKP1-1A* gene. (**d**) NpPP2-B10 interacted with NpSKP1-1A in vitro. (**e**) NpPP2-B10 interacted with NpSKP1-1A in vivo. Scale bars = 50 µm.

**Figure 6 ijms-24-07353-f006:**
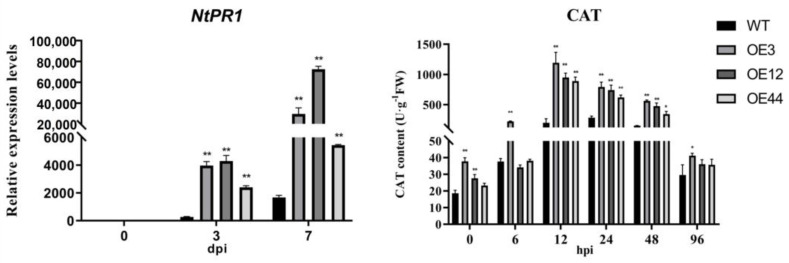
The expression of *NtPR1* and the activity of CAT in tobacco after infection with *P. nicotianae* race 0. Asterisks denote significant differences (compared with the control group): * *p* < 0.05, ** *p* < 0.01, by Student’s *t*-test.

**Figure 7 ijms-24-07353-f007:**
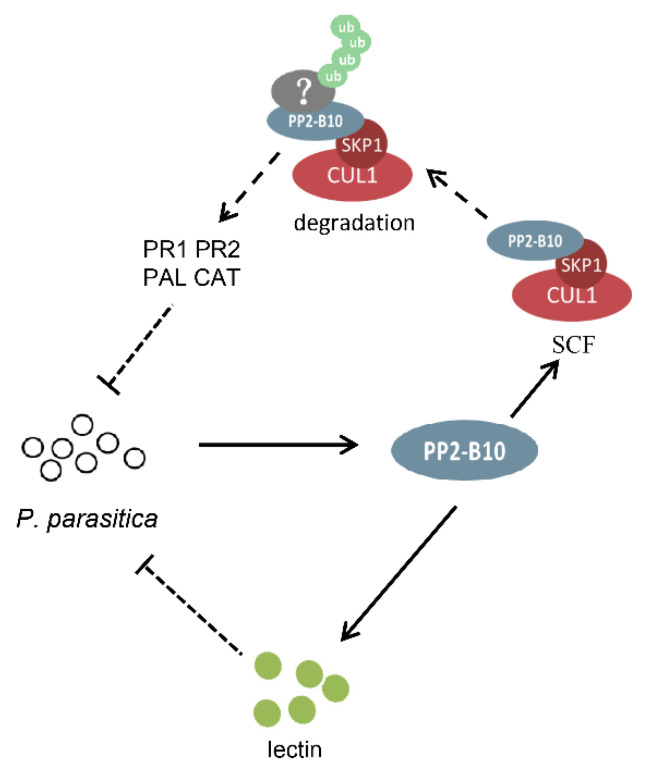
Model of the NpPP2-B10-mediated positive regulation of the plant defense response. The solid line represents our experimental results, and the dotted line represents our presumed pathway.

## Data Availability

The datasets presented in this study can be found in online repositories. The names of the repository/repositories and accession number(s) are: National Center for Biotechnology Information (NCBI) BioProject database, under accession numbers OM264751, OM264752, and OM264753.

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
