# Peer review of "NpPP2-B10, an F-Box-Nictaba Gene, Promotes Plant Growth and Resistance to Black Shank Disease Incited by Phytophthora nicotianae in Nicotiana tabacum"

_ijms, 2023, doi:10.3390/ijms24087353_

Round 1

Reviewer 1 Report

The Authors characterized a Nctaba-related lectin gene in Nicotiana plumbaginifolia, a wild tobacco species, which confers resistance to black shank disease caused by the oomycete Phytophthora nicotianae.

I read with interest the article which is written in a good English style. The Introduction is essential but updated (I added just a reference: please change the citation order of subsequent references). The experimental design and methods are appropriate. Results are presented clearly and discussed consequentially. 

I made just minor text editings. 

Two nomenclatural suggestions: 

- The updated name of the causal agent of tobacco black shank is Phytophthora nicotianae (some Authors still use Phytophthora parasitica var. nicotianae)

- Although the term Fungi is still used in a broad sense including  both true fungi and oomycetes, Phytophthora is an oomycete

Author Response

Dear Editor,

Please find attached our revision (“NpPP2-B10, an F-box-Nictaba gene, promotes plant growth and resistance to black shank disease incited by Phytophthora nicotianae in Nicotiana tabacum”) of a manuscript submitted to International Journal of Molecular Sciences (ijms-2333905). We thank you and the reviewers for the comments about our paper. With regard to these points, we have made substantial revisions.

Comments and Suggestions for Authors

The Authors characterized a Nctaba-related lectin gene in Nicotiana plumbaginifolia, a wild tobacco species, which confers resistance to black shank disease caused by the oomycete Phytophthora nicotianae.

  1. I read with interest the article which is written in a good English style. The Introduction is essential but updated (I added just a reference: please change the citation order of subsequent references). The experimental design and methods are appropriate. Results are presented clearly and discussed consequentially.

Response: Thank you very much for your valuable comments. We have added this reference [17] in line 610 and the corresponding content in line 71 (“Cell wall glycoproteins named cellulose-binding elicitor lectin (CBEL) have been found localized in the cell wall of P. nicotianae mycelium and play a key role as effectors during the plant-pathogen recognition process [17]. [17] La Spada F.; Stracquadanio C.; Riolo M.; Pane, A.;  Cacciola S.O. Trichoderma Counteracts the Challenge of Phytophthora nicotianae Infections on Tomato by Modulating Plant Defense Mechanisms and the Expression of Crinkler, Necrosis-Inducing Phytophthora Protein 1, and Cellulose-Binding Elicitor Lectin Pathogenic Effectors. Front. Plant Sci. 2020, 11, 583539. doi: 10.3389/fpls.2020.583539”).

  1. I made just minor text editings.

Response: We are very sorry for these error in the manuscript. We have revised them one by one in revised manuscript according to the annotated PDF file, including the title, abstract, introduction, results and discussion.

Two nomenclatural suggestions:

  1. The updated name of the causal agent of tobacco black shank is Phytophthora nicotianae (some Authors still use Phytophthora parasitica var. nicotianae)

Response: We have changed all "Phytophthora parasitica var. nicotianae)" to "Phytophthora nicotianae" in revised manuscript.

  1. Although the term Fungi is still used in a broad sense including both true fungi and oomycetes, Phytophthora is an oomycete

Response: We have changed "a fungal disease caused by Phytophthora" to "a disease caused by the oomycete Phytophthora" in line 388.

Reviewer 2 Report

I have just few mistakes and comments for this work:

Abstract:

- Not recommended to use abbreviations in the abstract (COD, POD) without the full naming first. In the case of SKP1, not sure

Introduction:

- I would recommend connecting better the phrases to enhance the impact of the assessments

- Line 57: 'and produce colloids' is repeated 

- Line 100:F-box- Nictaba is separated

Results:

- Line 122-123: be careful with the spaces and punctuation signs

Discussion:

- Line 346: mistake in Nctaba

- First part of the discussion is more or less a resume. Highly recommended, don't repeat these concepts and contextualize them in the scientific field

- In general, this section has a lack of citations to support or compare the results

Author Response

Dear Editor,

Please find attached our revision (“NpPP2-B10, an F-box-Nictaba gene, promotes plant growth and resistance to black shank disease incited by Phytophthora nicotianae in Nicotiana tabacum”) of a manuscript submitted to International Journal of Molecular Sciences (ijms-2333905). We thank you and the reviewers for the comments about our paper. With regard to these points, we have made substantial revisions.

Comments and Suggestions for Authors

I have just few mistakes and comments for this work:

  1. Abstract:

Not recommended to use abbreviations in the abstract (COD, POD) without the full naming first. In the case of SKP1, not sure

Response: We have changed "CAT and POD" to "enzymes catalase and peroxidase" in line 23.

  1. Introduction:

- I would recommend connecting better the phrases to enhance the impact of the assessments

- Line 57: 'and produce colloids' is repeated

Response: We have deleted "produce colloids" in line 57.

- Line 100:F-box- Nictaba is separated

Response: We have changed italic of " F-box-Nictaba" in line 100.

  1. Results:

- Line 122-123: be careful with the spaces and punctuation signs

Response: We have added or deleted spaces and punctuation in line 122-123.

  1. Discussion:

- Line 346: mistake in Nctaba

Response: We have changed " Nctaba" to " Nictaba" in line 346.

- First part of the discussion is more or less a resume. Highly recommended, don't repeat these concepts and contextualize them in the scientific field.

- In general, this section has a lack of citations to support or compare the results

Response: Thank you very much for your comments. We have rewritten first part of the discussion, mainly by cutting out repetitive concepts and results and adding comparable references.
